# Stomatal Regulation and Osmotic Adjustment in Sorghum in Response to Salinity

Pablo Rugero Magalhães Dourado [1], Edivan Rodrigues de Souza [1,*], Monaliza Alves dos Santos [1], Cintia Maria Teixeira Lins [1], Danilo Rodrigues Monteiro [1], Martha Katharinne Silva Souza Paulino [1] and Bruce Schaffer [2]

[1] Agronomy Department, Av. Dom Manuel de Medeiros, Dois Irmãos, Recife CEP 52171-900, Brazil; rugerodm@hotmail.com (P.R.M.D.); alves.monaliza@yahoo.com.br (M.A.d.S.); cintia_lins2@hotmail.com (C.M.T.L.); danilor.monteiro1@gmail.com (D.R.M.); marthakatharinne@gmail.com (M.K.S.S.P.)

[2] Tropical Research and Education Center, Department of Horticultural Sciences, Institute of Food and Agricultural Sciences, University of Florida, 18905 S.W. 280 Street, Homestead, FL 33031, USA; bas56@ufl.edu

* Correspondence: edivan.rodrigues@ufrpe.br

**Abstract:** *Sorghum bicolor* (L.) Moench, one of the most important dryland cereal crops, is moderately tolerant of soil salinity, a rapidly increasing agricultural problem due to inappropriate irrigation management and salt water intrusion into crop lands as a result of climate change. The mechanisms for sorghum's tolerance of high soil salinity have not been elucidated. This study tested whether sorghum plants adapt to salinity stress via stomatal regulation or osmotic adjustment. Sorghum plants were treated with one of seven concentrations of NaCl (0, 20, 40, 60, 80, or 100 mM). Leaf gas exchange (net $CO_2$ assimilation (A), transpiration (Tr); stomatal conductance of water vapor (gs), intrinsic water use efficiency (WUE)), and water ($\Psi w$), osmotic ($\Psi o$), and turgor $\Psi t$ potentials were evaluated at 40 days after the imposition of salinity treatments. Plants exhibited decreased A, gs, and Tr with increasing salinity, whereas WUE was not affected by NaCl treatment. Additionally, plants exhibited osmotic adjustment to increasing salinity. Thus, sorghum appears to adapt to high soil salinity via both osmotic adjustment and stomatal regulation.

**Keywords:** stomatal conductance; transpiration; net $CO_2$ assimilation; water and osmotic potentials; salt tolerance

## 1. Introduction

*Sorghum bicolor* (L.) Moench, one of the most important dryland cereal crops [1], is used for food, animal feed, and fuel. In addition to its resistance to water stress [2–4], this species with a $C_4$ photosynthetic pathway, is moderately tolerant to saline soil conditions, and therefore has the potential for cultivation in areas prone to salt water intrusion or high salinity of the irrigation water [5,6].

Soil salinity negatively affects the productivity of agricultural crops, hindering plant development through osmotic and ionic effects [7–9]. The adverse effects caused by soil salinity range from metabolic changes, ionic toxicity, and osmotic stress to biochemical and physiological disturbances [10]. Osmotic stress, as a result of a plant's exposure to salinity, has an immediate negative impact on water and nutrient absorption due to stomatal closure, which not only limits transpiration, but also inhibits photosynthesis [11–13]. High soil salinity also causes reductions in the leaf water potential ($\Psi w$), which further reduces osmotic ($\Psi o$) and turgor ($\Psi t$) potentials, hindering many physiological processes, and causing the accumulation of toxic ions and an increase in the amount of reactive oxygen species (ROS) in exposed plants [13,14].

Salt tolerance in sorghum, as in other crops, is not due to one trait but involves several traits including morphological, physiological, biochemical, and molecular markers [15].

These include maintenance of ionic homeostasis, transport and ion uptake, osmotic adjustment, and production of antioxidant enzymes [16]. Among these coping mechanisms, one of the most common is osmotic adjustment, which is characterized by the synthesis of compatible osmolytes that stabilize the structure of cells and proteins, maintaining the osmotic potential of the cell under osmotic stress [17,18]. Some of the salinity tolerance mechanisms reported in *Sorghum bicolor*, include proline accumulation, protection of photosynthetic enzymes and antioxidants [19–21], increased root hydraulic conductance [11], retention of plant water status, maintenance of the photosynthetic rate; increased concentrations of phenolic compounds [10] and turning on of genes associated with the detection and signaling and transport of Na+ in salt-specific QTL [6].

Stomatal conductance is often negatively impacted by soil salinity levels [22,23]. The low soil–water potential imposed by salinity can cause a marked decline in stomatal conductance (gs); the physiological rationale behind this reduction is the plant's attempt to minimize water loss under conditions of reduced water availability ("physiological drought") imposed by salinity. This reduction in gs often results in a reduction in net $CO_2$ assimilation, and therefore a reduction in plant growth [24]. To better understand the adaptive response of sorghum to high soil salinity, it is important to understanding the relative contribution of stomata and the relative cost to $CO_2$ assimilation and growth by determining stomatal conductance and net $CO_2$ assimilation [25].

Drought and salinity are two major abiotic stresses that severely limit agricultural production worldwide [26]. Plant response to salinity follows a biphasic model, wherein an early phase shows a similarity to drought (osmotic stress), and in the long term induces ion toxicity [27]. In response to drought stress, plants are classified as either isohydric, whereby plants reduce stress by closing their stomata, or anisohydric, whereby plants osmotically adjust to stress. Sorghum is classified as anisohydric because it adapts to drought stress by osmotic adjustment [28]. The objectives of this study were to determine if sorghum adapts to salinity stress in a similar manner as it does to drought stress via osmotic adjustment, or is stomatal regulation involved. Our hypothesis was that moderate tolerance of sorghum to soil salinity is solely due to osmotic adjustment. To test this hypothesis, we exposed sorghum plants to increasing soil salinity concentrations and we measured leaf gas exchange and osmotic adjustment at different salinity levels.

## 2. Materials and Methods

### 2.1. Experimental Design and Treatments

The experiment was conducted for 55 days in a greenhouse at the Federal Rural University of Pernambuco, Recife, Brazil. During the experiment, the average temperature and relative humidity in the greenhouse were 28.59 °C and 70%, respectively. Sorghum seeds (cv. IPA 2502) were sown in 10-L cylindrical plastic pots filled with Fluvic Neosol (Fluvisol) non-saline soil obtained from Pesqueira, Pernambuco, Brazil (8°34′11″ lat. and 37°48′54″ long). Initial soil chemical characteristics are shown in Table 1. Treatments consisted of irrigating plants with different salinity levels by adding differing concentrations of NaCl to the irrigation water beginning 15 after planting. Treatments were: 0, 10, 20, 40, 60, 80, or 100 mmol $L^{-1}$ of NaCl. The experiment was arranged as a randomized complete block design with seven treatments (salinity levels) and five single-plant replicates per treatment.

The bulk density, particle density, soil total porosity, sand, silt, and clay were 1.37 mg $m^{-3}$, 2.63 mg $m^{-3}$, 47.91 %, 433 g $kg^{-1}$, 466 g $kg^{-1}$, and 101 g $kg^{-1}$, respectively. The soil was maintained at 65% field capacity, at a moisture content of 0.19 g $g^{-1}$, equivalent to a matric potential of −0.01 Mpa (field capacity). Water lost by evapotranspiration was measured daily by weighing each pot in late afternoon. Each plant was then irrigated to bring each pot to 65% field capacity.

**Table 1.** Chemical characteristics of the of the Fluvic Neosol (Fluvisol) soil used in this study.

| Exchangeable Complex | Mean Value * | Soil Solution | Mean Value * |
|---|---|---|---|
| pH $_{(1:2.5)}$ | 6.75 | ECse (dS m$^{-1}$) | 3.36 |
| Ca$^{2+}$ (cmol$_c$ kg$^{-1}$) | 4.35 | Ca$^{2+}$ (mmol L$^{-1}$) | 9.12 |
| Mg$^{2+}$ (cmol$_c$ kg$^{-1}$) | 3.14 | Mg$^{2+}$ (mmol L$^{-1}$) | 8.63 |
| Na$^+$ (cmol$_c$ kg$^{-1}$) | 1.65 | Na$^+$ (mmol L$^{-1}$) | 13.51 |
| K$^+$ (cmol$_c$ kg$^{-1}$) | 1.20 | K$^+$ (mmol L$^{-1}$) | 2.13 |
| H$^+$ (cmol$_c$ kg$^{-1}$) | 1.54 | Cl$^-$ (mmol L$^{-1}$) | 25.47 |
| ESP (%) | 15.96 | SAR [(mmoles L$^{-1}$) $^{0.5}$] | 4.54 |

ESP: exchangeable sodium Percentage; ECse: electrical conductivity of the saturation paste extract; SAR: sodium adsorption relation Data are expressed as means. * $n$ = 10 samples.

### 2.2. Osmotic Potential, Water Potential, Turgor Potential, and Osmotic Adjustment

Fifty-five days after sowing (DAS) or forty days after the imposition of salinity treatments, five leaflets were collected from fully expanded leaves in the middle third of the canopy of each plant. Leaf water potential ($\Psi w$) was determined in each leaf sample with a Scholander pressure chamber (Model 1515D; PMS Instrument Company, Albany, OR, USA). The osmotic potential ($\Psi o$) in the leaf was quantified after freezing the same leaf sample used for $\Psi w$ determination, then thawing it and extracting the cell sap by macerating the leaf and filtering the extract through fine nylon mesh with the aid of a syringe. A drop of cell sap was placed on a filter paper disc and $\Psi o$ was measured with a vapor pressure osmometer (Vapro model 5600, Wescor, Inc., Logan, UT, USA). Osmometer readings (mmol kg$^{-1}$) were converted to -MPa and $\Psi o$ was calculated using the Van't Hoff equation [29]:

$$\Psi o = -RTC \tag{1}$$

where $C$ is the solute concentration; $R$ is the gas constant; and $T$ is the absolute temperature. The turgor potential ($\Psi t$) was calculated as the difference between $\Psi o$ and $\Psi w$. The osmotic adjustment ability was defined as the net increase in the solute concentration when the leaf was fully turgid in plants treated with NaCl compared to plants in the control treatment [30] and calculated by the equation:

$$OA = \Psi oc^{100} - \Psi os^{100} \tag{2}$$

where OA is the total osmotic adjustment, $\Psi oc^{100}$ is the osmotic potential of the control plants at full turgor and $\Psi os^{100}$ is the osmotic potential of the stressed plants at full turgor.

### 2.3. Leaf Gas Exchange

Leaf gas exchange (net CO$_2$ assimilation (A), transpiration (Tr), and stomatal conductance of water vapor (gs)) was measured 40 days after the imposition of salinity treatments, between 09:00 and 14:00 h in the first fully expanded leaf below the apex of the canopy. Leaf gas exchange was measured with a portable gas exchange system (model LI-6400XT, LI-COR Biosciences, Lincoln, NE, USA). In the leaf cuvette, the light intensity was maintained at 1800 mmol mol$^{-1}$, the ambient CO$_2$ concentration at 400 µmol mol$^{-1}$, and the air temperature at 25 °C. Intrinsic water use efficiency (WUE) was calculated as A/Tr.

### 2.4. Leaf Fresh Weight, Leaf Dry Weight, and Plant Height

At 40 days after the imposition of salinity treatments (55 days after sowing) plant height was measured and the leaves were collected for fresh and dry weight determinations. Leaves were oven dried at 65 °C prior to dry weight determination.

### 2.5. Statistical Analyses

The data were analyzed by linear regression using R Statistical Software [31].

## 3. Results

### 3.1. Xylem Osmotic, Water, and Turgor Potentials, and Osmotic Adjustment Ability

There was a strong inverse linear relationship between $\Psi w$ ($R^2$ = 0.99) or $\Psi o$ ($R^2$ = 0.99) and NaCl concentration (Figure 1). The $\Psi w$ decreased from −0.10 MPa in the control treatment to −0.90 MPa in the 100 mM treatment, and the $\Psi o$ decreased from −0.80 MPa in the control treatment to −1.5 MPa in the 100 mM NaCl treatment (Figure 1). Although there was also a significant inverse linear relationship between $\Psi t$ and NaCl concentration ($R^2$ = 0.84), the decrease was more gradual than for $\Psi w$ or $\Psi o$ as indicated by a lower slope of the regression line (Figure 1) for $\Psi t$ versus NaCl concentration compared to the $\Psi w$ or $\Psi o$ regression lines. After 55 days, electrical conductivity of saturated paste extracts from each treatment was 2.9, 5.4, 9, 16.4, 20.7, 24.6, and 33.8 dS m$^{-1}$ for the 0, 10, 20, 40, 60, 80, and 100 mM of NaCl treatments, respectively.

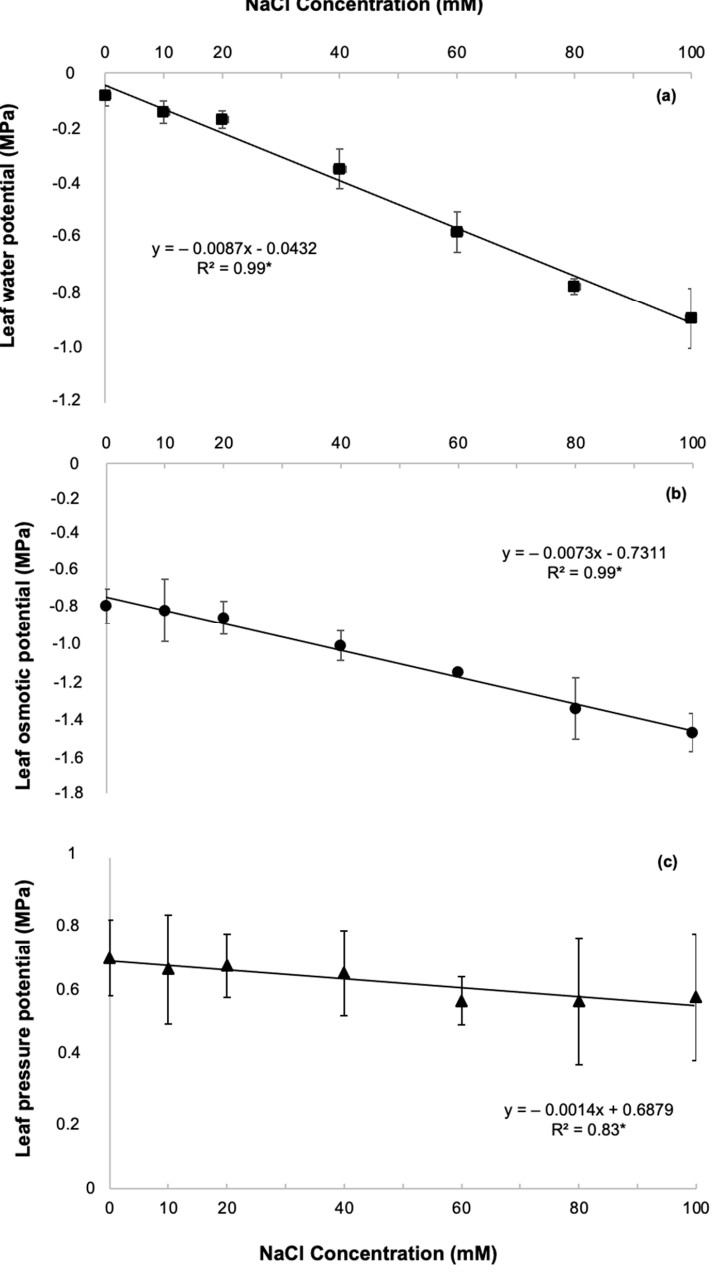

**Figure 1.** (**a**) Water, (**b**) osmotic, and (**c**) turgor potentials in sorghum leaves, 40 days after NaCl treatments were imposed. Symbols represent the means of each treatment and error bars indicate ± 1 std. dev. * ($p < 0.05$).

There was a strong positive linear relationship ($R^2$ = 0.97) between NaCl concentration and OA (Figure 2). The osmotic adjustment increased from 0.23 in the control treatment to 0.7 in the 100 mM NaCl treatment.

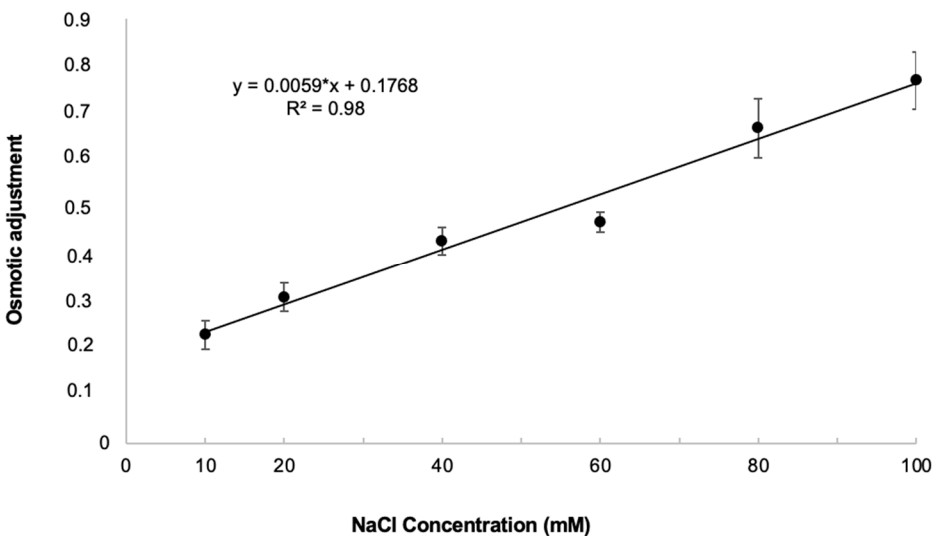

**Figure 2.** Osmotic adjustment in sorghum plants 40 days after NaCl treatments were imposed. Symbols represent means of each treatment. Symbols represent the means of each treatment and error bars indicate ± 1 std. dev. * ($p < 0.05$).

*3.2. Leaf Gas Exchange*

There was a strong inverse linearly relationship ($R^2$ = 0.97) between NaCl concentration and A (Figure 3). Net $CO_2$ assimilation decreased by 0.204 per 1 mM of increase in NaCl concentration.

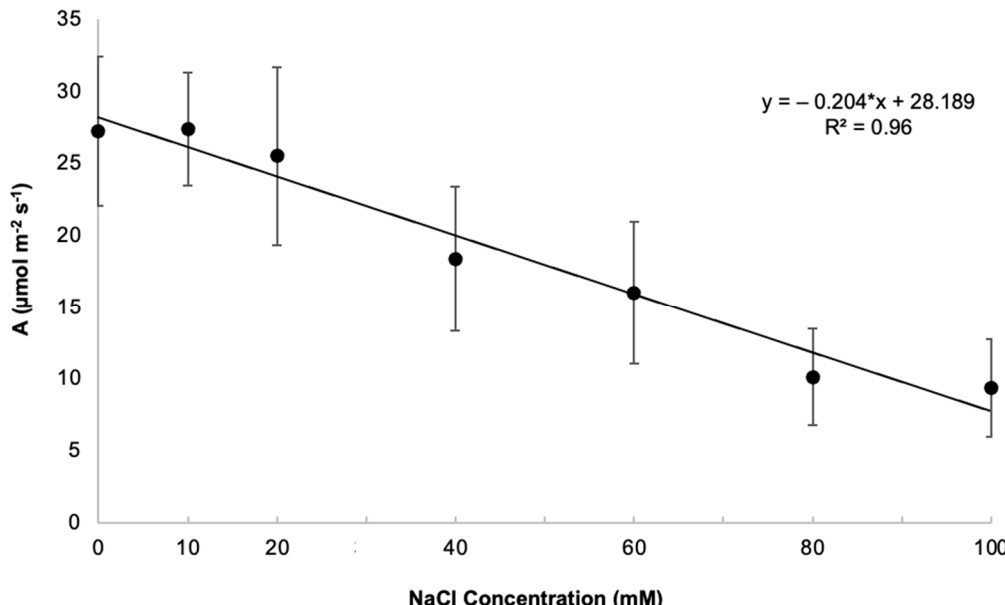

**Figure 3.** Net $CO_2$ assimilation (A) of sorghum plants 40 days after NaCL treatments were imposed. Symbols represent means of each treatment. Symbols represent the means of each treatment and error bars indicate ± 1 std. dev. * ($p < 0.05$).

Similar to A, there was a significant linear decrease in gs ($R^2$ = 0.95) and Tr ($R^2$ = 0.97) as NaCl concentration increased, whereas WUE was not affect by NaCl concentration and was similar for all treatments (Figure 4).

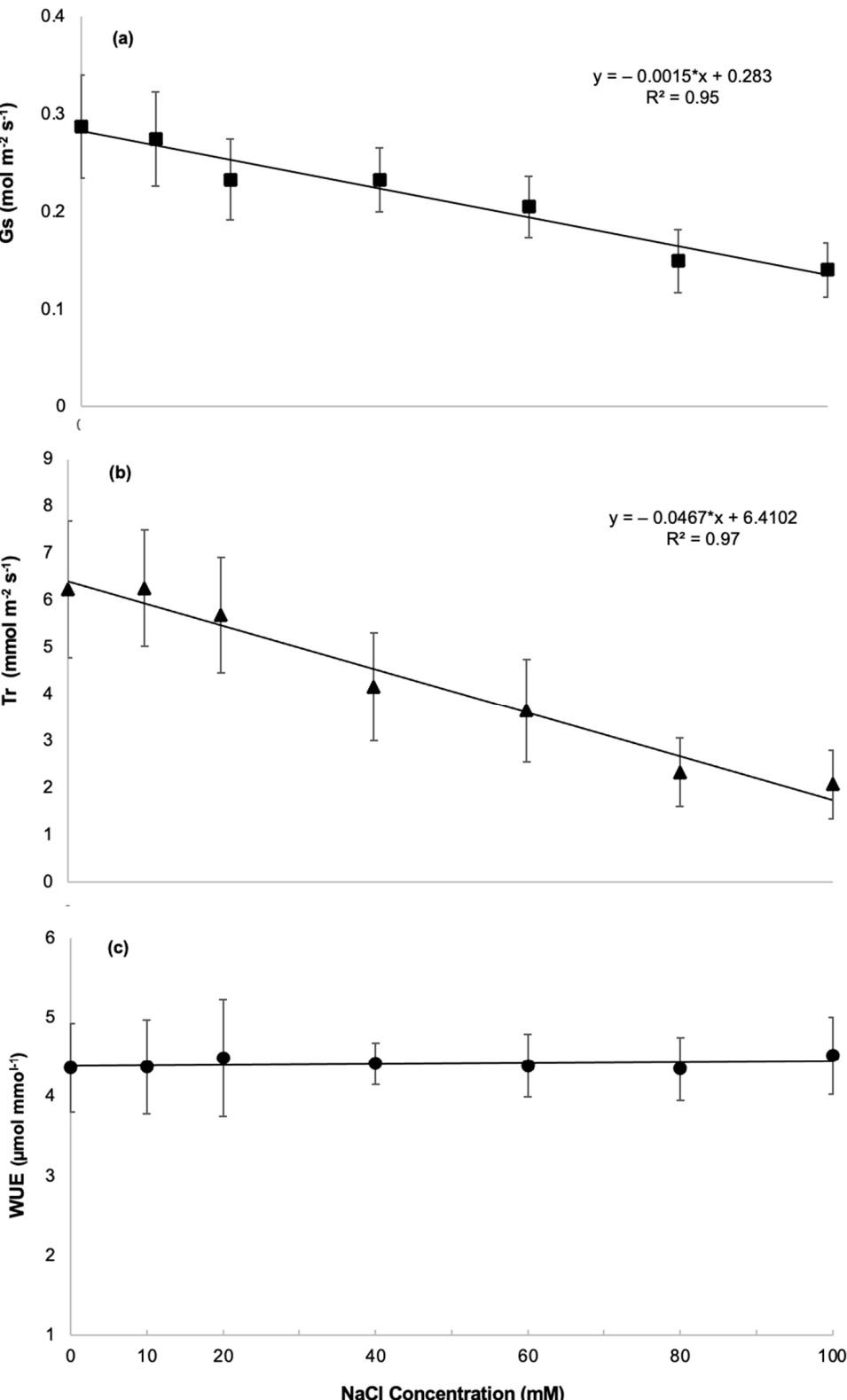

**Figure 4.** (**a**) Stomatal conductance (gs), (**b**) transpiration, and (**c**) and intrinsic water use efficiency (WUE) (D) of sorghum seedlings, 40 days after NaCl treatments were imposed. Symbols represent means of each treatment. Symbols represent the means of each treatment and error bars indicate ± 1 std. dev. * ($p < 0.05$).

There was a strong linear correlation between A ($R^2 = 0.88$) or Tr ($R^2 = 0.89$) and gs (Figure 5). For both variables, plants in the control and lower NaCl treatments were grouped at the top of the regression line and plants in the highest NaCl treatments grouped at the bottom of the regression line (Figure 5), indicating that A and Tr decreased as a result of decreased gs in response to increasing soil salinity.

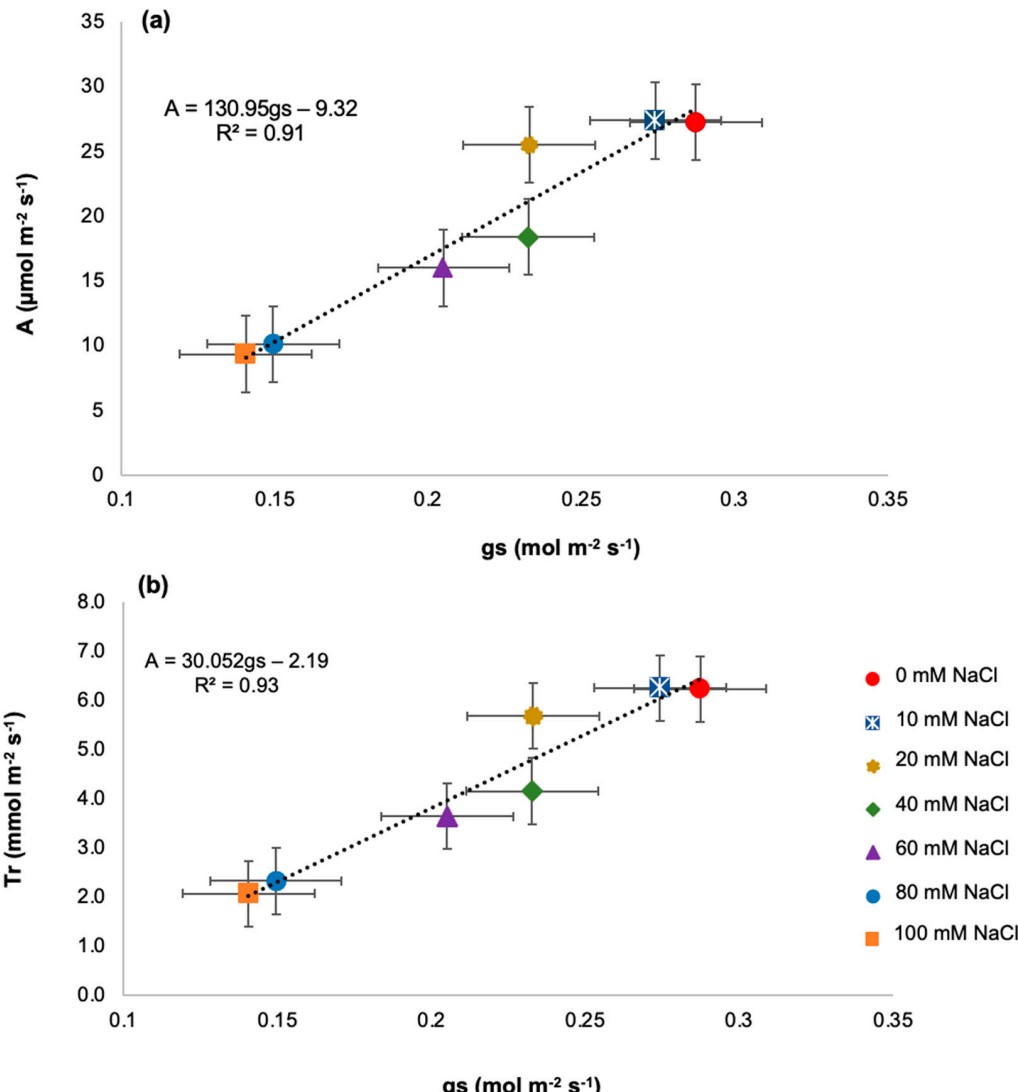

**Figure 5.** (**a**) Relationship between on a net $CO_2$ assimilation (A) and stomatal conductance (gs), and (**b**) transpiration (Tr) and (gs) in sorghum plants in different NaCl treatments, 40 days after NaCl treatments were imposed. Symbols represent the means of each treatment and error bars indicate ± 1 std. dev.

### 3.3. Leaf Fresh Weight, Leaf Dry Weight, and Plant Height

There was a linear decrease in plant height, leaf fresh weight, and leaf dry weight as salinity increased (Figure 6). For the highest salinity treatment (100 mM of NaCl) the reductions in plant height, leaf fresh weight, and leaf dry weight were 27% (154 cm to 112 cm), 48% (99 to 51 g plant$^{-1}$), and 33% (18 to 12 g plant$^{-1}$), respectively, compared to the control treatment (0 mM of NaCl).

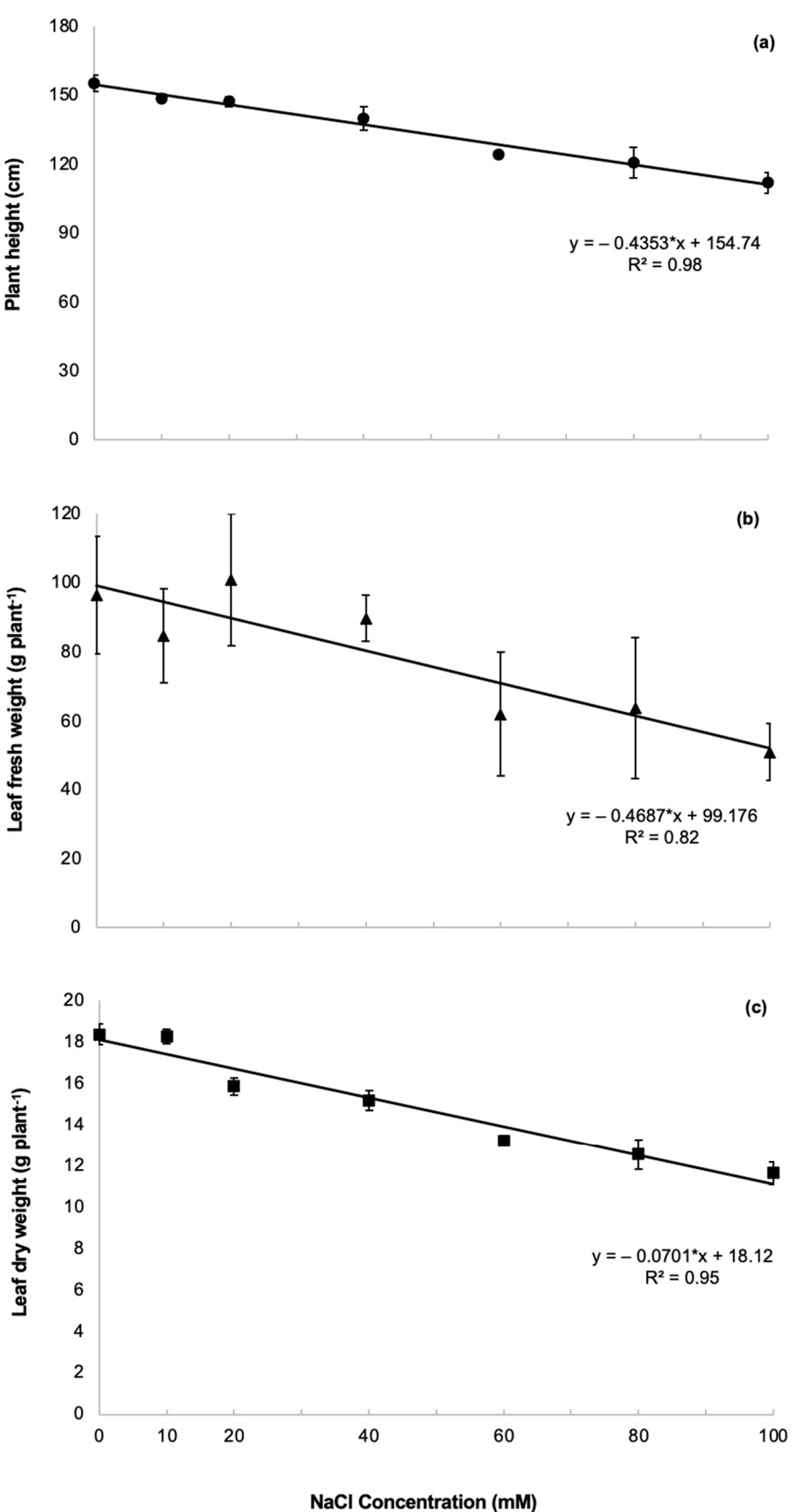

**Figure 6.** (**a**) Plant height, (**b**) leaf fresh weight, and (**c**) leaf dry weight in sorghum plants in different NaCl treatments, 40 days after NaCl treatments were imposed. Symbols represent the means of each treatment and error bars indicate ± 1 std. dev. * ($p < 0.05$).

## 4. Discussion

The observation that there was less of a decrease in $\Psi t$ with increasing NaCl concentration compared $\Psi o$ or $\Psi w$ suggests that there is the capacity for osmotic adjustment in sorghum. This was confirmed by OA measurements, which indicated that the values of $\Psi w$, $\Psi o$, and $\Psi t$ could be used to assess osmotic adjustment in the absence of direct determinations of OA. Monteiro et al. [32] evaluated the same cultivar of sorghum evaluated in the present study and found $\Psi w$ values ranged from $-0.119$ MPa (0 dS m$^{-1}$) to $-0.875$ MPa (7.5 dS m$^{-1}$ – 75 mM of NaCl), which were similar to values observed in the present study. In saline soil conditions, many plants osmotically adjust by accumulating solutes which function to regulate $\Psi o$ or $\Psi w$, allowing plants to maintain water uptake and/or $\Psi t$ [33], thereby decreasing stress. Inorganic solutes, such as potassium, magnesium, chloride, and nitrate have all been shown to contribute to as much as 52% of osmotic adjustment in sorghum plants, while organic solutes contribute to approximately 30% of the osmotic adjustment [34]. In a study of different varieties of sorghum, Bafeel [35] suggested that sorghum plants survive in saline conditions due to the osmotic adjustment involving accumulation of inorganic salts in the vacuole and accumulation of organic solutes in the cytoplasm. Negrão, Schmöckel, and Tester [8] observed compartmentalization of toxic ions into specific tissues, cells, and subcellular organelles as one the key strategies of plant adaptation to salt stress. A similar situation may occur in sorghum plants under high salinity conditions.

Lacerda et al. [36] tested the quantitative and qualitative aspects of leaf and root osmotic adjustment in two genotypes of sorghum cultivated in NaCl concentrations of 0, 50 and 100 mmol L$^{-1}$. Our results from leaf osmotic potential in plants treatment with 0 and 100 mmol L$^{-1}$ of NaCl ($-0.77$ and $-1.47$ MPa, respectively) were similar to the values found in the previous study for the salt-tolerant genotype ($-0.752$ and $-1.204$ MPa for 0 and 100 mmol L$^{-1}$ treatments, respectively). It is important to note that the genotype (IPA 2502) we tested is recommended for semiarid regions affected by abiotic stress such as salinity and drought in northeastern Brazilian. A relevant discussion about increasing osmotic adjustment is related to the balance of Na and Cl versus compatible solutes. According to Lacerda et al. [36], the higher decrease in $\Psi s$ in the salt sensitive genotype was due to a higher Na$^+$ and Cl$^-$ accumulation and suggested the importance of evaluating the osmotic adjustment quality.

The decrease in A with increasing NaCl concentrations observed in the present study was also observed by Nabati et al. [37], who found that after 21 days of exposure of sorghum to high NaCl concentrations (electrical conductivity of 10.5 and 23.1 dS m$^{-1}$), A decreased by 18 and 26%, respectively, compared to a treatment with a lower electrical conductivity of 5.2 dS m$^{-1}$ ($-52$ mM NaCl). The negative effect of high salinity on A is related to a decrease in the osmotic potential of the soil solution, which limits water uptake by the roots [38], resulting in stomatal closure to conserve water. As a result of stomatal closure, gs and Tr are reduced and there is a limitation of CO$_2$ diffusion into the leaf thereby limiting A [39]. This is supported by our observation that the concomitant reductions of A and Tr offset each other, resulting in no significant effect of NaCl concentration on WUE. Plants growing in saline soils often adapt to high salinity by minimizing water loss because growth depends on the ability to maintain A, while reducing water loss [8]. This was not the case with sorghum. Although we observed that sorghum was able to maintain WUE when gs was reduced, the decrease in A at high salinity levels inhibited plants growth under high soil salinity (Figure 6). This may partially explain why sorghum is considered only moderately tolerant of saline soil conditions.

Salinity effects on photosynthesis are often associated with inhibition of electron transport proteins in chloroplasts [40–42]. Wang et al. [43] determined that in response to high soil salinity, there was a reduction in a complex of three proteins in *Ricinus communis* that negatively influenced the initial stage of CO$_2$ fixation, compromising CO$_2$ uptake and fixation dye to decreased by Rubp-carboxylase/oygenase (RuBisCO) activity. Thus, the decreasing A in sorghum as salinity increased in the present study may have not only

been due to physical factors such as changes in water potentials, by may also have been affected by biochemical factors such as reduced enzyme activity. It was also reported that several photosynthetic proteins involved in the stability of PSII and photosynthetic electron transport from photosystem II to photosystem I are affected by salt stress [42]. The authors also observed that a NaCl-induced reduction in enzymes involved in the Calvin cycle and the first step of carbon fixation, such as carbonate dehydratases, are potentially regulated by salinity stress.

Calone et al. [44] compared the growth of sorghum to three salinity levels (0, 3, or 6 dS m$^{-1}$) with leaching (water applied to above water holding capacity, of the soil) and without leaching (irrigated below water holding capacity of the soil). When comparing the 0 to the 6 dS m$^{-1}$ treatments, they observed an 87% and 42% reduction in dry weight without and with leaching, respectively. In our study, where there was no leaching, and plant growth differences between highest salinity level and the control treatment were less than those observed by Calone et al. [44]. Growth differences between the present study and those observed by Calone et al. [44] may have been due to the difference in physical and/or chemical qualities of the soils, source of salt and/or genotype tested, which have a significant impact on results [25]. The present study provides new information about salinity effects on an important sorghum genotype that is grown commercially in areas of Northeast Brazil that are prone to high salinity levels.

## 5. Conclusions

Our hypothesis that sorghum's ability to moderately tolerate high soil salinity is due to osmotic adjustment (similar to their tolerance to drought stress), was only partially true. Our data showed that the sorghum plants respond to increasing soil salinity by both osmotic adjustment and by stomatal regulation, as indicated by reductions in gs with increasing salt concentrations. However, there was a metabolic cost when soil salinity was high due to A being limited by reduced gs under these conditions. The concomitant decreased in Tr with decreasing A as soil salinity increased, resulted in maintenance of WUE even at high salinity, allowing sorghum to tolerate high soil salinity (100 mM of NaCl = −10 dS m$^{-1}$). Sorghum is not a halophytic species and soil salinity reached 33.8 dS m$^{-1}$ at 55 days after exposure to the 100 mM of NaCl treatment, suggesting salinity tolerance, which is also supported by low growth reductions, such as only 27% and 33% to plant height and dry weight, respectively, in the 100 mM of NaCl treatment compared to the control treatment.

**Author Contributions:** Conceptualization, P.R.M.D. and E.R.d.S.; methodology-investigation, P.R.M.D., E.R.d.S., M.A.d.S., C.M.T.L., D.R.M. and M.K.S.S.P.; writing—original draft preparation, P.R.M.D., E.R.d.S., M.K.S.S.P., B.S.; writing—review and editing, P.R.M.D., E.R.d.S., M.A.d.S. and B.S.; funding acquisition, E.R.d.S., B.S. All authors have read and agreed to the published version of the manuscript.

**Funding:** This research was funded by the National Council for the Improvement of Higher Education (CAPES) and the National Council for Scientific and Technological Development for granting scholarships and funding the research project (CNPq n° 308530/2015-2 and 420723/2016-1) and Foundation for Science and Technology Development of the State of Pernambuco.

**Institutional Review Board Statement:** Not applicable.

**Informed Consent Statement:** Not applicable.

**Data Availability Statement:** Available upon request.

**Conflicts of Interest:** The authors declare no conflict of interest.

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
