# Peer review of "Stomatal Regulation and Osmotic Adjustment in Sorghum in Response to Salinity"

_agriculture, doi:10.3390/agriculture12050658_

Round 1
Reviewer 1 Report
The Authors took into account all my suggestions. In reference 25, the journal name is bold, not the year.
Reviewer 2 Report
The author has made the changes.
Reviewer 3 Report
Authors add some clarification in their writings, which is acceptable
Best
This manuscript is a resubmission of an earlier submission. The following is a list of the peer review reports and author responses from that submission.
Round 1
Reviewer 1 Report
The work presented for review describes the current problem of adapting plants to salt stress, which is more and more common due to the climate changes. The work concerns an economically important plant, grown in regions particularly exposed to salinity. The paper presents relatively few results, but enough for this type of article (communication). The authors used the latest literature in the paper, about 65% of the literature is from the last 5 years, which increases the scientific value of the work. However, I have some comments regarding methodology and edition.
1) some measurments were made on plants 55 days after the introduction of salinity stress, and some after 40 days - it should be explained what is the reason for adopting such an experimental model
2) it should be clarified at what point the seedlings (p. 2 line 72) were planted into the soil (e.g. 7 days after imbibition) or maybe the seeds were sown?
3) in my opinion, in the first figure, each graph should have a scale on the x-axis - it would be easier to interpret
4) in the description to the Figure 1, there is no explanation of which graph is marked as a) b) c)
5) in the descriptions of Figures 4 and 5 it is indicated that the results were obtained 45 days after the introduction of the salt stress, and in the methods paragragraph is mentioned that the measurements were performed after 40 days
6) in Figure 5, the graphs are not described a) and b), and there is such a designation in the description
7) in the graphs in Figures 1-4, bars indicating the standard deviation should be added
8) the sentence on page 9 lines 228-230 should be redrafted, because in such version it is not precise
9) p. 9 line 225: after “the stomatal”, “closure” should be added
10) p. 9 line 244: after “observed”, “that” should be added
11) p. 1 line 1 article type shoud be changed into Communication
Author Response
Ref.: No. agriculture-1641225
Stomatal Regulation and Osmotic Adjustment in Sorghum in Response to Salinity
Dear Reviewer
We are submitting a revised manuscript (changes are highlighted in green). We have followed all the suggestions and recommendations
REVIEWER 01
- The work presented for review describes the current problem of adapting plants to salt stress, which is more and more common due to the climate changes. The work concerns an economically important plant, grown in regions particularly exposed to salinity. The paper presents relatively few results, but enough for this type of article (communication). The authors used the latest literature in the paper, about 65% of the literature is from the last 5 years, which increases the scientific value of the work. However, I have some comments regarding methodology and edition.
Authors’ Response: We appreciate the reviewer’s comments and have incorporated the suggestions into the revised manuscript.
some measurments were made on plants 55 days after the introduction of salinity stress, and some after 40 days - it should be explained what is the reason for adopting such an experimental model
Authors’ Response: We clarified this information in the manuscript. The experiment was conducted for 55 days. The salinity treatment started 15 days after planting sorghum in pots. All the measurements (water potential and leaf gas exchange) were made 55 days after planting and 40 days after the imposition of salinity treatments.
It should be clarified at what point the seedlings (p. 2 line 72) were planted into the soil (e.g. 7 days after imbibition) or maybe the seeds were sown?
Authors’ Response: We corrected this information. We stated seed instead seedling.
In my opinion, in the first figure, each graph should have a scale on the x-axis - it would be easier to interpret
Authors’ Response: We adjusted the x-axis as suggested.
In the description to the Figure 1, there is no explanation of which graph is marked as a) b) c)
Authors’ Response: We added the explanations in the caption for Figure 1.
In the descriptions of Figures 4 and 5 it is indicated that the results were obtained 45 days after the introduction of the salt stress, and in the methods paragraph is mentioned that the measurements were performed after 40 days
Authors’ Response: The information was corrected and adjusted to 40 days after the introduction of salinity treatments.
In Figure 5, the graphs are not described a) and b), and there is such a designation in the description
Authors’ Response: We described the designations.
In the graphs in Figures 1-4, bars indicating the standard deviation should be added
Authors’ Response: We added standard deviation bars to all figures.
The sentence on page 9 lines 228-230 should be redrafted, because in such version it is not precise
Authors’ Response: We redrafted the text as suggested.
- 9 line 225: after “the stomatal”, “closure” should be added
Authors’ Response: We added it
- 9 line 244: after “observed”, “that” should be added
Authors’ Response: We added it.
- 1 line 1 article type should be changed into Communication
Authors’ Response: We changed it.

Reviewer 2 Report
- The main problem is that there are many mistakes in writing, which need rewriting.
- Line 30: fodder, animal feed. Is it repeated?
- Line 46: Reference labels are inconsistent.
- Please provide tables of the original data set and data analysis process. It's important.
- Please provide photos of salt treatment in the greenhouse.
- There is too little information about stomata's involvement in salt stress in the preface and discussion section. Please rewrite it.
Author Response
Ref.: No. agriculture-1641225
Stomatal Regulation and Osmotic Adjustment in Sorghum in Response to Salinity
Dear Reviewer
We are submitting a revised manuscript (changes are highlighted in green). We have followed all the suggestions and recommendations
REVIEWER 02
The main problem is that there are many mistakes in writing, which need rewriting.
Authors’ Response: The entire revised version of the manuscript was thoroughly reviewed by one of the co-authors, Dr. B. Schaffer, who is a Professor at the University of Florida and native English speaker. He assured all co- authors that the English grammar is correct.
Line 30: fodder, animal feed. Is it repeated?
Authors’ Response: We removed fodder
Please provide tables of the original data set and data analysis process. It's important.
Authors’ Response: We presented the results after checking the statistics. In addition, we added the error bars indicating the standard deviation for all figures as suggested by Reviewer 1. We believe that provides a sufficient indication of the range and variability of the data.
Please provide photos of salt treatment in the greenhouse.
Authors’ Response: Unfortunately, we don’t have photos comparing the treatments
There is too little information about stomata's involvement in salt stress in the preface and discussion section. Please rewrite it.
Authors’ Response: We added a new paragraph in the introduction, as follows:
Stomatal conductance is often negatively impacted by soil [22, 23]. The low soil-water potential imposed by salinity can cause a marked decline in stomatal conductance (gs); the physiological rationale behind this reduction is the plant's attempt to minimize water loss under conditions of reduced water availability (“physiological drought”) imposed by salinity. This reduction in gs often results in a reduction in net CO2 assimilation, and therefore a reduction in plant growth [24]. To better understand the adaptive response of sorghum to high soil salinity, it is important to understanding the relative contribution of stomata and the relative cost to CO2 assimilation and growth by determining stomatal conductance and net CO2 assimilation [25].
References added:
- Soltabayeva,A.; Ongaltay, A.; Omondi, J.O.; Srivastava, S. Morphological, Physiological and Molecular Markers for Salt-Stressed Plants. Plants 2021, 10, 243, doi: 10.3390/plants10020243
- Rasouli, F.; Kiani-Pouya, A.; Shabala, L.; Li, L.; Tahir, A.; Yu, M.; Hedrich, R.; Chen, Z.; Wilson, R.; Zhang, H.; et al. Salinity Effects on Guard Cell Proteome in Chenopodium quinoa. Int. J. Mol. Sci. 2021, 22, 428, doi: 10.3390/ijms2201 0428
- Zhao, C.;Zhang, H.; Song, C.; Zhu, J-K,; Shabala,S. Mechanisms of Plant Responses and Adaptation to Soil Salinity. The Inno-vation. 2020, 1, 1, doi: 10.1016/j.xinn.2020.100017
- Amombo, E.; Ashilenje, D.; Hirich, A.; Kouisni, L.; Oukarroum, A.; Ghoulam, C.; Gharous, M. E.; Nilahyane, A. Exploring the correlation between salt tolerance and yield: research advances and perspectives for salt‐tolerant forage sorghum selection and genetic improvement. Planta 2022, 255:71, doi: 10.1007/s00425-022-03847-w

Reviewer 3 Report
This work focused on salt-stressed Sorghum plants, especially measuring their Leaf gas exchange parameters, and xylem water, osmotic and turgor potentials under different salt conditions. However, these physiological parameters were commonly changed under salt stress in different plants. In this work, there is no mechanism found for osmotic adjustment, or molecular mechanism. And it is better to use the controls positive which have high osmotic adjustment and stomatal regulation, and negative control as with low osmotic adjustment and stomatal regulation plants, then the claim will be more reliable
The measured physiological parameters in Sorghum under salinity stress were the common parameters which will be changed under salinity stress in other plants, please read Soltabayeva et al 2021 and/or any review articles related with salinity stress. The authors did not give the mechanism for this process, they needed to go forward to molecular markers in Sorghum under salinity stress and give some mechanism for the observed response of sorghum. Second, they claim that Sorghum exhibited osmotic adjustment and stomatal regulation to increase salinity, where they needed to use control plants which are known to regulate their growth on salinity stress through osmotic adjustment and stomatal regulation and also negative plants which show the opposite.
I hope they improve their information
Author Response
Ref.: No. agriculture-1641225
Stomatal Regulation and Osmotic Adjustment in Sorghum in Response to Salinity
Dear Reviewer
We are submitting a revised manuscript
REVIEWER 03
This work focused on salt-stressed Sorghum plants, especially measuring their Leaf gas exchange parameters, and xylem water, osmotic and turgor potentials under different salt conditions. However, these physiological parameters were commonly changed under salt stress in different plants. In this work, there is no mechanism found for osmotic adjustment, or molecular mechanism. And it is better to use the controls positive which have high osmotic adjustment and stomatal regulation, and negative control as with low osmotic adjustment and stomatal regulation plants, then the claim will be more reliable
The measured physiological parameters in Sorghum under salinity stress were the common parameters which will be changed under salinity stress in other plants, please read Soltabayeva et al 2021 and/or any review articles related with salinity stress. The authors did not give the mechanism for this process, they needed to go forward to molecular markers in Sorghum under salinity stress and give some mechanism for the observed response of sorghum. Second, they claim that Sorghum exhibited osmotic adjustment and stomatal regulation to increase salinity, where they needed to use control plants which are known to regulate their growth on salinity stress through osmotic adjustment and stomatal regulation and also negative plants which show the opposite.
Authors’ Response: We thank the reviewer for the suggestions. It is true that we only tested one cultivar. However, as previously stated in our comment to another reviewer, the response to salinity of this commercially important cultivar, commonly planted in saline soils has not been previously evaluated. Also, in a previous study (Lacerda et al., 2003) compared osmotic adjustment in two other sorghum cultivars and found that both exhibited osmotic adjustment to salinity but there was a difference in the quality of the adjustment. We have now referenced that article. However, that study did not also investigate stomatal or photosynthetic responses. As stated in our hypothesis, our goal was not to compare osmotic adjustment between sorghum cultivars or to determine the molecular mechanisms involved in osmotic adjustment, but to determine the relative contribution of stomatal versus osmotic adjustment to soil salinity in this commercially important cultivar that is often planted in soils from semiarid regions and affected by salinity and drought
